# *Rhodiola rosea* Reduces Intercellular Signaling in *Campylobacter jejuni*

**DOI:** 10.3390/antibiotics11091220

**Published:** 2022-09-08

**Authors:** Ajda Kunčič, Franz Bucar, Sonja Smole Možina

**Affiliations:** 1Department of Food Science and Technology, Biotechnical Faculty, University of Ljubljana, Jamnikarjeva ulica 101, 1000 Ljubljana, Slovenia; 2Department of Pharmacognosy, Institute of Pharmaceutical Sciences, University of Graz, Beethovenstraße 8, 8010 Graz, Austria

**Keywords:** *Rhodiola rosea*, proanthocyanidins, flavonoids, *Campylobacter jejuni*, LuxS, intercellular signaling

## Abstract

*Campylobacter jejuni* is a major foodborne pathogen and the leading cause of bacterial gastroenteritis, i.e., campylobacteriosis. Besides searching for novel antimicrobials, identification of new targets for their action is becoming increasingly important. *Rhodiola rosea* has long been used in traditional medicine. Ethanolic extracts from the roots and rhizomes of the plant contain a wide range of bioactive compounds with various pharmacological activities. In this study, cultivated plant materials have been used, i.e., “Mattmark” and “Rosavine”. Through optimized protocols, we obtained fractions of the initial ethanolic extracts rich in most important bioactive compounds from *R. rosea*, including salidroside, rosavins, proanthocyanidins (PACs), and flavonoids. The antimicrobial activity in relation to the chemical composition of the extracts and their fractions was studied with an emphasis on *C. jejuni* AI-2-mediated intercellular signaling. At concentration 15.625 mg/L, bioluminescence reduction rates varied from 27% to 72%, and the membrane remained intact. Fractions rich in PACs had the strongest antimicrobial effect against *C. jejuni*, with the lowest minimal inhibitory concentrations (MICs) (M F3 40%: 62.5 mg/L; R F3 40%: 250 mg/L) and the highest intercellular signaling reduction rates (M F3 40%: 72%; R F3 40%: 65%). On the other hand, fractions without PACs were less effective (MICs: M F5 PVP: 250 mg/L; R F5 PVP: 1000 mg/L and bioluminescence reduction rates: M F5 PVP: 27%; R F5 PVP: 43%). Additionally, fractions rich in flavonoids had strong antimicrobial activity (MICs: M F4 70%: 125 mg/L; R F4 70%: 250 mg/L and bioluminescence reduction rates: M F4 70%: 68%; R F4 70%: 50%). We conclude that PACs and flavonoids are crucial compound groups responsible for the antimicrobial activity of *R. rosea* roots and rhizomes in *C. jejuni*.

## 1. Introduction

Overuse of antibiotics in veterinary and human medicine is responsible for the development of antibiotic-resistant strains. Due to increased antimicrobial resistance, the search for novel antimicrobial agents and their potential targets is of utmost importance. Antibiotic resistance has also been recorded for *Campylobacter jejuni*, a major foodborne pathogen and the leading cause of bacterial gastroenteritis, also known as campylobacteriosis. This infection occurs very commonly in European countries. With over 120,000 confirmed human cases of illness in 2020, it has been the most commonly reported zoonosis, representing over 60% of all reported cases in 2020 [1].

There are different potential targets that can be affected by antimicrobials. Among them are bacterial intercellular communication systems, collectively termed quorum sensing (QS) [2]. QS is a bacterial cell-to-cell communication and it refers to the ability of bacteria to sense information from other cells in the population [3]. Bacteria communicate through small molecules also known as autoinducers (AIs) [4]. The autoinducer-2 (AI-2) signaling molecule regulates interspecies communication as it is present in both Gram-positive and Gram-negative bacteria [5,6]. Elvers and Park [7] described the role of the *luxS* gene in the production of the AI-2 signal and AI-2-mediated signaling in *C. jejuni*. The LuxS enzyme has a central role as a metabolic enzyme in the methyl cycle, responsible for the generation of S-adenosyl-L-methionine (SAM). The enzyme has its role in the two-step reaction of homocysteine formation, where AI-2 is formed as a by-product [8]. As LuxS-deficient *C. jejuni* shows critical differences in colonization and virulence [9], AI-2-mediated signaling can be considered as a potential target for the control of that pathogen.

Natural compounds with anti-QS potential can be found in medicinal plants and their extracts [10]. Various phytochemicals inhibit *C. jejuni* intercellular signaling. Examples of this include citrus ethanolic extracts [11], epigallocatechin gallate (EGCG) [12], resveratrol inclusion complexes [13], coriander essential oil, linalool [14], *Euodia ruticarpa* ethanolic extract [15], carvacrol [16], curcumin, allyl sulfide, garlic oil, and ginger oil [17]. Šimunović et al. [9] investigated the correlation of QS inhibition with changes in *C. jejuni* motility, adhesion to polystyrene surfaces, and adhesion to and invasion of INT407 cells. A positive correlation was reported between *C. jejuni* QS reduction and reduced motility, adhesion to polystyrene surfaces, and invasion. In their screening, among 20 natural extracts, essential oils, and pure compounds, *Rhodiola rosea* ethanolic extract showed the best overall antimicrobial activity against *C. jejuni*.

*Rhodiola rosea* L. (*Sedum roseum* (L.) Scop.) is an herbaceous plant belonging to the family Crassulaceae. Vernacular names by which we can recognize the plant include roseroot, golden root, and arctic root [18]. The plant grows up to 70 cm in height and has fleshy, succulent leaves. The flowers form a compact whorled inflorescence on top of the halms. The root system forms thick rhizomes [19].

*R. rosea* has long been used in traditional medicine. A key factor determining the quality of its formulations is the quantification of salidroside, rosavin, and rosarin [20]. In Northern Europe, Russia, and North America, *R. rosea* extracts are standardized to contain at least 3% rosavins and from 0.8% to 1% salidroside [21,22]. The majority of these extracts are derived from wild plants harvested in Russia and Mongolia, which threatens the long-term conservation of natural populations. In addition, fraudulent material containing non-*R. rosea* plant material is suspected to be on the market. In order to preserve the natural sources of *R. rosea* plant material (the collection of which is prohibited in many countries), and to ensure the quality and authenticity of the plant material, domestication and cultivation of the plant seems to be the most appropriate solution [23].

Extracts of the plant’s underground organs contain various compounds, including monoterpene alcohols and their glycosides, cyanogenic glycosides, aryl glycosides, phenylethanoids (salidroside, p-tyrosol), phenylpropanoids and their glycosides (rosavins), flavonoids, flavonolignans, proanthocyanidins (PACs), and gallic acid derivatives [24]. According to Ma et al. [25], salidroside, p-tyrosol, rosarin, rosavin, rosin, and rosiridin are responsible for the biological activity of *R. rosea*. Besides being used in pharmaceutical preparations, the plant is popular as a food additive and dietary supplement. The plant’s underground organs can be used as a crude drug (dried and powdered) or as an extract [26].

Due to its wide range of biologically active compounds, the plant has various pharmacological activities. Among them are antioxidative, anti-inflammatory, anti-fatigue, anti-depressive, anxiolytic, anti-tumor, antimicrobial, neuroprotective, cardioprotective, and normalizing endocrine and immune activities [24,27,28,29].

In the present study, ethanolic extracts were prepared from *R. rosea* cultivated plant material, i.e., “Mattmark” and “Rosavine”, of which “Mattmark” is the first synthetic *R. rosea* cultivar [23]. For the first time, biologically active compounds or compound groups from *R. rosea* were separated into different fractions, and antimicrobial activity, with an emphasis on AI-2-mediated intercellular signaling in relation to the chemical composition of the extracts and their fractions, was studied. Due to great antimicrobial activity having been previously reported for the ethanolic extract from *R. rosea* in *C. jejuni*, the aim of the study was to evaluate whether certain compounds or compound groups are crucial for this activity.

## 2. Material and Methods

### 2.1. Bacterial Strains and Growth Conditions

*C. jejuni* NCTC 11168, *C. jejuni* 11168∆*luxS*, and *Vibrio harveyi* MM30 are part of Chair of Biotechnology, Microbiology and Food Safety (Biotechnical Faculty, University of Ljubljana, Ljubljana, Slovenia) strain collection and were stored at −80 °C. *C. jejuni* strains were stored in a 20% glycerol (Kemika, Zagreb, Croatia) and an 80% Muller-Hinton broth (MHB) (Oxoid, Hampshire, UK) solution. *V. harveyi* MM30 was stored in a 20% glycerol and an 80% AI bioassay (AB) liquid medium solution (composed as described in [9]). A reference strain, *C. jejuni* NCTC 11168, was revitalized by cultivating on Karmali selective media (Oxoid, Hampshire, UK), followed by cultivating on Mueller–Hinton agar (MHA) (Oxoid, Hampshire, UK) at 42 °C in microaerophilic conditions (85% N_2_, 10% CO_2_ and 5% O_2_). A mutant strain, *C. jejuni* 11168∆*luxS*, which does not produce the AI-2 signal [2], was cultivated on MHA with the addition of 30 mg/L kanamycin (Merck, Darmstadt, Germany). A reporter strain, *V. harveyi* MM30, with eliminated AI-2 production [30], was used for AI-2 bioassays and was directly inoculated into 5 mL of AB liquid medium and incubated at 30 °C in aerobic conditions.

### 2.2. Plant Material

*R. rosea* roots and rhizomes from cultivated plant material were provided by Dr. Christoph Carlen and Mr. Claude-Alain Carron (Agroscope, Conthey, Switzerland). Two types of plant material were used: “Mattmark”, harvested in April 2017 in Conthey at an altitude of 460 m, and “Rosavine”, harvested in June 2012 in Bruson at an altitude of 1050 m. Voucher specimens were kept at the Institute of Pharmaceutical Sciences, Department of Pharmacognosy, University of Graz.

### 2.3. Extracts Preparation

Ethanolic extracts were prepared from the roots and rhizomes of the plant. Sliced dried plant material was ground and about 50 g was used for the extraction. The lipophilic compounds were removed by pre-extraction with hexane (Carl Roth, Karlsruhe, Germany). Ethanolic extraction with 96% ethanol (Carl Roth, Karlsruhe, Germany) took place in an ultrasonic bath (Elma, Singen, Germany) for 10 min at room temperature, followed by centrifugation at 4000 RPM for 10 min. Ethanolic extraction was repeated three times, the supernatants were collected, and the solvent was evaporated using a rotary evaporator (Heidolph, Schwabach, Germany), followed by freeze drying using a VirTis Sentury freeze dryer (SP Scientific, Buena, CA, USA) resulting in 28.52% (“Mattmark”) and 32.47% (“Rosavine”) extraction yields. The ethanolic extracts, designated as M/R 96% EtOH, were stored in dark glass flasks in a fridge at 4 °C.

### 2.4. Fractionation of the Extracts: Fractions Preparation

Five fractions, enriched with different compounds or compound groups, were obtained from each crude ethanolic extract. Fractionation was performed on DIAION HP-20 adsorbent resin (Sigma-Aldrich, Steinheim, Germany) or using Polyclar AT (polyvinylpyrrolidone, or PVP) (Serva, Heidelberg, Germany). For the elution of compounds from the column, we used micropure water (MW; Barnstead Easypure RF) and different concentrations of methanol (Carl Roth, Karlsruhe, Germany).

The separation protocol described by Sun et al. [31] for the purification of salidroside and p-tyrosol was modified to obtain fractions of the ethanolic extracts as follows. DIAION HP-20 adsorbent resin was pre-treated by soaking it in 70% methanol/MW solution at 4 °C overnight, followed by washing with MW in a Buchner funnel until no alcohol remained. Then, 100 mg of crude ethanolic extract was dissolved in 10 mL of MW and centrifuged at 4000 RPM for 10 min at room temperature. The dissolved extract was mixed with 15 g of pre-treated DIAION HP-20 adsorbent resin. After full adsorption of substances to the adsorbent, it was loaded into a glass column and the bed volume (BV) was determined as 24 mL. Subsequently, the column was eluted with two BV of MW and two BV of a 20%, 40%, and 70% methanol/MW solution. The eluents were collected and their composition was determined directly by UHPLC-PDA-ESI-MS analysis. The optimized elution protocol was scaled up by a factor of 12 in order to obtain enough material for further microbiological analysis, resulting in fractions of M/R F1 0%, M/R F2 20%, M/R F3 40%, and M/R F4 70%.

The fifth fraction of each crude ethanolic extract was prepared using PVP, which mainly binds compounds with phenolic OH groups, e.g., tannins, PACs, and many flavonoids. Therefore, 50 mg of crude ethanolic extract was dissolved in 15 mL of 50% methanol/MW solution, followed by centrifugation at 4000 RPM for 10 min at room temperature. The supernatant was added to 1.25 g of PVP, mixed, and again centrifuged at 4000 RPM for 10 min at room temperature. The supernatant was filtered through a filter paper (Carl Roth, Karlsruhe, Germany) and the solvent was evaporated using a rotary evaporator, followed by freeze drying. In this case, the optimized protocol was scaled up by a factor of 20 to obtain enough material for further microbiological analysis, resulting in fractions M/R F5 PVP.

### 2.5. Chemical Characterization (UHPLC-PDA-ESI-MS Analysis)

Analysis of crude extracts and their fractions was carried out on two Dionex UltiMate 3000 UHPLC systems (Thermo Fisher Scientific, Waltham, CA, USA). The first system was coupled to a linear ion-trap mass spectrometer (MS), LTQ XL, equipped with an electrospray ionization (ESI) ion source (Thermo Fisher Scientific, Waltham, California, USA). Both systems were equipped with a photodiode array detector (PDA) (Thermo Fisher Scientific, Waltham, California, USA). Separation was undertaken on a Zorbax SB-C18 rapid resolution HD column (Agilent, Santa Clara, California, USA), 100 × 2.1 mm, 1.8 µm particle size. The mobile phase consisted of MW + 0.1% formic acid (A) (Honeywell Fluka, Seelze, Germany) and acetonitrile (VWR International, Rosny-sous-Bois-cedex, France) + 0.1% formic acid (B). A gradient elution was performed, starting with 2% B, increasing to 22% B at 13.33 min, 70% B at 22.22 min, then dropping back to 2% B at 22.67 min and keeping this composition until the end (28 min). The column temperature was set to 40 °C. Flow rate was set to 0.450 mL/min. Injection volume was 2 μL. Samples of crude ethanolic extracts were prepared using 20% ethanol/MW (2:8) in a concentration of 5 mg/mL and centrifuged at 13,000 RPM for 10 min at room temperature before analysis. Fractions were prepared using MW and different concentrations of ethanol/MW or methanol/MW. Therefore, we analyzed the fractions directly, after centrifugation at 13,000 RPM for 10 min at room temperature. PDA detection was performed in the 190 nm to 500 nm wavelength range. The mass spectra were recorded in negative and positive ion mode in the m/z range of 50 to 2000 amu. Mass spectral conditions were set as follows: source voltage 5.0 kV (ESI positive), 3.5 kV (ESI negative); capillary temperature 350 °C; source temperature 300 °C; sheath gas flow 40 arb, auxiliary gas flow 10 arb.

### 2.6. Determination of Minimal Inhibitory Concentrations (MICs)

The MICs of preparations against *C. jejuni* NCTC 11168 and *C. jejuni* 11168∆*luxS* were determined by the broth microdilution method previously described [32] with some modifications. Stock solutions were prepared in dimethyl sulfoxide (DMSO) (Merck, Germany) and later diluted in MHB to a final concentration of 1000 mg/L and 2.5% of DMSO. The bacterial inoculum contained approximately 5 × 10^5^ colony-forming units (CFU)/mL. Cell viability was determined using resazurin fluorescent dye solution, prepared as previously described [9]. The fluorescence intensity was measured using the microplate reader Varioskan LUX (Thermo Fischer Scientific, Waltham, CA, USA) at an excitation wavelength of 560 nm and an emission wavelength of 590 nm. The MIC represents the lowest concentration where fluorescence, expressed in relative fluorescence units (RFU), reached the value of the negative control. The MICs were determined in triplicate.

### 2.7. AI-2 Bioassay

AI-2 bioassays were performed to evaluate the influence of the preparations on *C. jejuni* AI-2-mediated intercellular signaling. The cell-free spent media (SMs) were prepared and AI-2 bioassays were performed as described in [9]. Briefly, *C. jejuni* NCTC 11168 (*C. j.* 11168-Tr.) and *C. jejuni* 11168∆*luxS* (*C. j.* 11168∆*luxS*-Tr.) cultures with approximately 5 × 10^5^ CFU/mL were treated separately with sub-inhibitory concentrations of preparations (i.e., 0.25× MIC or lower). Stock solutions were prepared in DMSO and were further diluted in MHB to a final 1% of DMSO. After 24 h incubation, SMs were prepared by filter-sterilization through 0.2 µm syringe filters (Sartorius, Göttingen, Germany). The SMs were stored in a freezer at −20 °C.

For AI-2 bioassays, *V. harveyi* MM30 was used as a reporter strain because it does not produce the AI-2 signal itself [30,33]. The bacterial culture was prepared with approximately 5 × 10^3^ CFU/mL. Bioluminescence was measured using a microplate reader (Varioskan LUX) at 30 min intervals over 22.5 h. An impact of preparations in sub-inhibitory concentrations on *C. jejuni* intercellular signaling was determined indirectly from the reported bioluminescence of *V. harveyi* MM30 expressed in relative light units (RLU) and compared with the non-treated control (*C. j.* 11168-C+). To confirm that LuxS-deficient *C. jejuni* does not produce an AI-2 signal, we added *V. harveyi* MM30 culture to 5% of *C. jejuni* 11168∆*luxS* (*C. j.* 11168∆*luxS*-C+) SM and measured the bioluminescence of the reporter strain. By measuring the bioluminescence of *V. harveyi* MM30 without the addition of *C. jejuni* SM, we proved that the strain does not produce the AI-2 signal itself. Bioluminescence reduction rates (%) were calculated using equation 1. They indicate how *C. jejuni* AI-2-mediated intercellular signaling is reduced due to the exposure of bacteria to the preparations in comparison with the non-treated control.
(1)$$\mathrm{Bioluminescence}\;\mathrm{reduction}\;\mathrm{rate}\;\left[\%\right]=100-\left(\left(\frac{\left(C.\;j.\;11168-\mathrm{Tr}.-C.\;j.\;11168∆luxS-\mathrm{Tr}.\right)}{\left(C.\;j.\;11168-C+-C.\;j.\;11168∆luxS-C+\right)}\right)\times 100\right)$$

To examine whether compounds of the most active fractions (i.e., PACs from Fractions 3) bind to AI-2 signal molecule, additional AI-2 bioassays were performed. In this case, we added preparations in sub-inhibitory concentrations to the initial SMs, and after 1 h incubation we measured bioluminescence of the *V. harveyi* MM30 reporter strain. The signals were compared with the ones from the original SMs, and bioluminescence reduction rates were calculated again. All the experiments were carried out in triplicate.

### 2.8. Membrane Integrity

The influence of preparations on *C. jejuni* membrane integrity was determined using a LIVE/DEAD BacLight Bacterial Viability kit (L-7012; Molecular Probes, Eugene, CA, USA).

The membrane disruption assays were performed according to Kovač et al. [34]. A mixture of green fluorescent dye SYTO 9 and propidium iodide (PI) was prepared according to the manufacturer instructions. Stock solutions of the extracts and their fractions were prepared in DMSO at 100-fold higher concentrations and were further diluted in MHB to the final 1% of DMSO. The dye mixture was added to 100 μL of the treated or non-treated *C. jejune* cultures with approximately 5 × 10^5^ CFU/mL (1:1, *v*/*v*). The kinetics of PI intracellular penetration was measured in RFU by a microplate reader (Varioskan LUX) in terms of the SYTO 9 fluorescence at an excitation wavelength of 481 nm and an emission wavelength of 510 nm. Kinetic measurements over the last 10 min of the assays were used to calculate the membrane disruption (%). The experiment was carried out in triplicate.

### 2.9. Statistical Analysis

Before statistical tests, normality was tested for all the data using Shapiro–Wilk and Kolmogorov–Smirnov tests. Based on the normality test, we determined statistical significance between treatments and control using the Mann–Whitney U test in the case of non-normal distribution of the data, and using one-way ANOVA in case the data were normally distributed. At a 95% confidence interval, the results were statistically significant at value *p* < 0.05. The analyses were performed using IBM SPSS Statistics software, version 23 (IBM Corp., Armonk, CA, USA).

## 3. Results

### 3.1. Chemical Characterization of the Extracts and Fractions

The composition of the ethanolic extracts was qualitatively analyzed by comparing the retention time, mass spectrometric fragmentation, and UV spectra from UHPLC-PDA-ESI-MS data analysis with data from the literature [35,36,37,38,39]. Eighteen compounds were identified in the “Mattmark” ethanolic extract (Appendix A, Appendix A) and seventeen in the “Rosavine” ethanolic extract (Appendix A, Appendix A). Aside from typical salidroside (phenylethanoid) and rosavins (cinnamyl alcohol glycosides), chromatograms recorded at UV 270 nm indicated significant amounts of PACs as a complex mixture, of which only a few compounds could be assigned to epigallocatechin-3-O-gallate (EGCG), and two dimers containing EGCG and epigallocatechin (EGC), respectively. Two flavonoid glycosides could be assigned to the herbacetin-7-O-glycosides rhodiosin and rhodionin. The main difference between the “Mattmark” and “Rosavine” plant material is in the quantity of certain compounds or compound groups. Thus, “Mattmark” (Appendix A) contains more PACs, rosavins, and flavonoids than “Rosavine”. On the other hand, “Rosavine” (Appendix A) has higher amounts of salidroside, based on peak area comparison in UV chromatograms.

Five fractions were prepared from each crude ethanolic extract of the plant material. The main compounds or compound groups of each fraction are listed in Table 1. The composition of each fraction obtained from the “Mattmark” ethanolic extract is shown on Figure 1, and a comparison of peak areas of selected compounds is presented in Appendix A. By the adsorption of the crude ethanolic extracts to DIAION HP-20 adsorbent resin and the stepwise elution of compounds by increasing methanol concentrations, the separation of compound groups could be obtained, although salidroside could be found in Fractions F2 and Fractions F3, and rosavins could be found in Fractions F2 (trace amounts), Fractions F3, and Fractions F4. After adsorption of the crude ethanolic extracts to PVP, UHPLC analysis indicated that PACs and flavonoids, as well as gallic acid, were almost quantitatively removed from the extracts (Fractions F5).

### 3.2. Determination of MICs

The MICs were determined by the broth microdilution method for the reference strain *C. jejuni* NCTC 11168 and for the mutant strain *C. jejuni* 11168∆*luxS* (Table 2). Preparations with MICs under 1000 mg/L were tested further in sub-inhibitory concentrations, i.e., 0.25× MICs or lower, to avoid an impact on cell growth.

For *C. jejuni* NCTC 11168, the MIC of the “Mattmark” ethanolic extract was 125 mg/L, while the MICs of its fractions varied from 62.5 to 500 mg/L. The MIC of the “Rosavine” ethanolic extract was 500 mg/L, while the MICs of its fractions varied from over 1000 to 250 mg/L. In general, Fractions 3 and Fractions 4 showed better or the same antimicrobial activity than the ethanolic extracts, while Fractions 1, Fractions 2, and Fractions 5 were less or equally effective. The MICs of the “Mattmark” ethanolic extract and its fractions were lower compared with the MICs of the “Rosavine” ethanolic extract and its fractions.

### 3.3. Inhibition of Intercellular Signaling

AI-2 bioassays were performed to evaluate whether the ethanolic extracts and their fractions, in sub-inhibitory concentrations, affected *C. jejuni* AI-2-mediated intercellular signaling.

First, the impact of the ethanolic extracts and their fractions on AI-2 production in *C. jejuni* was tested in sub-inhibitory concentrations of 0.25× MICs. In this case, all the preparations significantly inhibited *C. jejuni* signaling (*p* < 0.05), with bioluminescence reduction rates from 54% to 91% (Figure 2A). The “Mattmark” ethanolic extract and its fractions were more effective in intercellular signaling reduction, although they were tested in lower concentrations than the “Rosavine” ethanolic extract and its fractions. Overall, *R. rosea* preparations showed a high impact on *C. jejuni* signaling but, due to the variation in tested concentrations, it was not possible to assign specific compounds or compound groups to the observed effect.

Consequently, the reduction in *C. jejuni* AI-2-mediated intercellular signaling by the preparations was tested at the same concentration, i.e., 15.625 mg/L, which corresponds to 0.25× MIC of “Mattmark” Fraction 3. By testing preparations at the same concentration, we aimed to find out if some extracts/fractions were more effective in reducing intercellular signaling. Both ethanolic extracts, as well as almost all their fractions, significantly affected *C. jejuni* signaling, even if present at such a low concentration (*p* < 0.05). Only “Mattmark” Fraction 5 did not have a significant impact (*p* > 0.05). Bioluminescence reduction rates varied from 27% to 72% (Figure 2B). Both Fractions 3 most effectively reduced *C. jejuni* signaling when tested at the same concentration. Additionally, Fractions 4 had a great impact. On the other hand, Fractions 5, which were devoid of PACs and flavonoids, were less effective (Figure 2B). 

### 3.4. Disruption of Membrane Integrity

Membrane integrity assays were performed to determine if the disruption of *C. jejuni* membrane is also involved in the antimicrobial activity of *R. rosea* preparations. As in previous experiments, preparations were first tested at sub-inhibitory concentrations of 0.25× MICs, followed by testing at the same concentration, i.e., 15.625 mg/L.

At 0.25× MICs, the disruptive impact on membrane integrity varied from 3% to 31% (Figure 3(A1,A2)). In this case, the disruption of membrane integrity is likely to contribute to the antimicrobial activity of at least some of the preparations. On the other hand, when tested at the same concentration of 15.625 mg/L, the disruptive impact on membrane integrity was very low and varied from 2% to 6% (Figure 3(B1,B2)). We assume that such an effect does not contribute to the antimicrobial activity of the preparations.

## 4. Discussion

QS represents an important mechanism for modulating *C. jejuni* behavior within its population [40]. In the present study, ethanolic extracts of cultivated plant material, i.e., “Mattmark” and “Rosavine”, were prepared and fractionated by optimized protocols in order to obtain five fractions rich in salidroside, rosavins, PACs, and/or flavonoids. The fifth fractions, with almost no gallic acid, PACs, or flavonoids, were prepared from the initial ethanolic extracts to determine whether this group of compounds could play the most important role in the antimicrobial activity of *R. rosea* ethanolic extract.

According to the MICs, Fractions 1—rich in mainly simple gallic acid (M/R F1 0%)—and Fractions 5—without gallic acid, PACs, or flavonoids (M/R F5 PVP)—have lower antimicrobial activity, corresponding to higher MICs than the ethanolic extracts (R/M 96% EtOH). Moreover, Fractions 5 have lower antimicrobial activity than Fractions 3 (M/R F3 40%), which are rich in PACs. This indicates that PACs, including the monomeric EGCG, could be one of the crucial compound groups responsible for the antimicrobial activity of *R. rosea* ethanolic extract. Furthermore, flavonoids (herbacetin glycosides) might also contribute to anti-*Campylobacter* activity when comparing Fractions 4 and Fractions 5.

AI-2 bioassays were performed to determine the effect of the ethanolic extracts and their fractions on *C. jejuni* AI-2-mediated signaling. At concentrations of 0.25× MICs, both ethanolic extracts and their fractions showed a statistically significant impact (*p* < 0.05), with bioluminescence reduction rates ranging from 54 to 91%. Due to the variation in tested concentrations of the preparations, it was not possible to refer to the compounds or groups of compounds that play the most important role in this. Therefore, the preparations were tested at the same concentration, i.e., 15.625 mg/L, which also represents the 0.25× MIC of M F3 40%. All preparations, except R F5 PVP, significantly affected *C. jejuni* signaling (*p* < 0.05), even at such a low concentration. Here, it is important to note that M F3 40% showed a greater reduction in intercellular signaling than R F3 40%. Besides this, M F5 PVP showed much lower impact than R F5 PVP. It is also important that “Mattmark” plant material contained a higher amount of PACs and flavonoid than “Rosavine” plant material. In addition, Fractions 3 were enriched in PACs, including EGCG, while Fractions 5 had the PACs removed. These results suggest that PACs may be one of the crucial compounds in *R. rosea* with the potential to reduce *C. jejuni* signaling. High bioluminescence reduction rates (from 65% to 72%) were achieved in the case of treating *C. jejuni* with a sub-inhibitory concentration of Fractions 3, compared with lower impact (from 27% to 43%) when *C. jejuni* was treated with sub-inhibitory concentration of Fractions 5. Because M F3 40% contains more PACs than R F3 40%, a greater loss of activity was observed for “Mattmark” when we removed those compounds, as we did in Fractions 5. Similarly to MIC determination, the herbacetin glycosides rhodionin and rhodiosin could also be relevant for intercellular signaling reduction. This can be deduced from the significant reduction in bioluminescence by Fractions 4 (which were rich in these flavonoids) and the decrease in the respective effects in Fractions 5 (which were also devoid of flavonoid glycosides). The stronger intercellular signaling reduction by M F4 70% compared with R F4 70%, showing a lower concentration of flavonoids, supports the assumption that the herbacetin glycosides contribute to the observed intercellular signaling reduction of the crude extract. A high QS reduction rate (>90%) had previously been reported when *C. jejuni* was treated with sub-inhibitory concentrations of, e.g., *C. limon* [11] and *R. rosea* [9] ethanolic extracts. Strong anti-QS activity of EGCG from green tea against *C. jejuni* NCTC 11168 was previously reported by the authors of [12]. In their case, EGCG decreased the bioluminescence reported for *V. harveyi* BB152 of 96% when tested at a sub-inhibitory concentration of 0.75× MBC or 65.25 mg/L. Our study supports the conclusion of Šimunović et al. [9] that QS or AI-2-mediated intercellular signaling is a potential target in the control of *C. jejuni* and that various natural plant-based preparations act as true intercellular signaling modulators. As an upgrade to their research, we can see from our results that PACs, including EGCG and the herbacetin glycosides, are the most important compound groups of *R. rosea* crude ethanolic extract responsible for AI-2-mediated signaling reduction in *C. jejuni*.

This study also demonstrated that the antimicrobial activity of the extracts and their fractions was due to the reduction in *C. jejuni* signaling without affecting the membrane integrity. Even though preparations at 0.25× MICs did not affect cell growth, the membrane integrity was significantly disrupted (*p* > 0.05) by “Rosavine” ethanolic extract and its Fractions 3 and Fractions 4. Nevertheless, no significant effect (*p* < 0.05) on membrane integrity was observed compared with the non-treated control when the extracts and their fractions were tested at a concentration of 15.625 mg/L. The study also excluded that PACs affect *C. jejuni* signaling by binding to the AI-2 signaling molecule which was released to the growth medium. The AI-2 bioassays have shown that the reported bioluminescence of the reporter strain *V. harveyi* MM30 did not vary significantly (*p* < 0.05) between the initial SMs and the SMs to which we once again added Fractions 3, rich in PACs, at a concentration of 15.625 mg/L. For example, the bioluminescence reduction rates varied from 1% to 2% between the initial SMs obtained after treating *C. jejuni* with sub-inhibitory concentrations of Fractions 3 and the same SMs to which preparations were again added in sub-inhibitory concentrations (data not shown).

PACs are the most abundant plant-derived polyphenols belonging to one of the tannin groups (condensed tannins). They are among the most commonly consumed dietary polyphenols. Condensed tannins are able to form insoluble complexes with carbohydrates and proteins [41]. Studies have shown that mucosal immunity to pathogen infection can be enhanced by PACs [42]. In addition, our study supports the fact that PACs, in this case from *R. rosea* underground organs, represent natural compounds with the ability to reduce intercellular signaling in order to fight *C. jejuni*. This is in agreement with a recent publication of Hao et al. [43]: this confirmed EGCG with significant inhibitory effects on the development of biofilm, protease, elastase activity, swimming, and swarming motility, which were also positively related to the production of C4-AHL signaling molecules in *Pseudomonas aeruginosa*. In addition, herbacetin glycosides of *R. rosea* might also contribute to *C. jejuni* signaling reduction. Interestingly, a study by the authors of [44] showed that herbacetin has a high affinity to LuxR-type protein *Shewanella baltica* in a virtual screening.

## 5. Conclusions

In this study, we provided a protocol for the separation of bioactive compounds or compound groups from *R. rosea* roots and rhizomes. Our results suggest PACs and flavonoids are the most important compound groups from *R. rosea*, with great potential for *C. jejuni* AI-2-mediated signaling reduction. Nevertheless, it is still unclear whether AI-2 in *C. jejuni* represents a true QS signaling molecule or whether it is a metabolic by-product of a crucial central metabolic methyl cycle pathway [45]. To date, AI-2 receptors have still not been identified [46], and mechanisms by which bioactive compounds affect *C. jejuni* signaling should be further investigated.

## Figures and Tables

**Figure 1 antibiotics-11-01220-f001:**
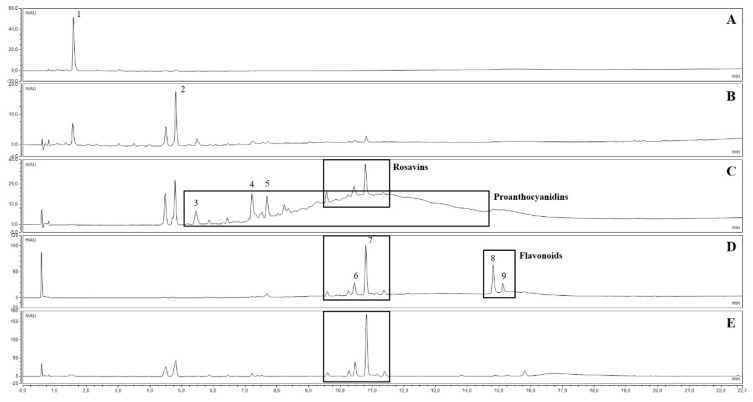
UV chromatograms at 270 nm of “Mattmark” fractions (**A**): Fraction 1 (M F1 0%); (**B**): Fraction 2 (M F2 20%); (**C**): Fraction 3 (M F3 40%); (**D**): Fraction 4 (M F4 70%); (**E**): Fraction 5 (M F5 PVP)). Compounds: 1 = gallic acid, 2 = salidroside, 3 = EGC-EGCG dimer, 4 = EGCG-EGCG dimer, 5 = EGCG, 6 = rosarin, 7 = rosavin, 8 = rhodiosin, 9 = rhodionin; for details see Appendix A.

**Figure 2 antibiotics-11-01220-f002:**
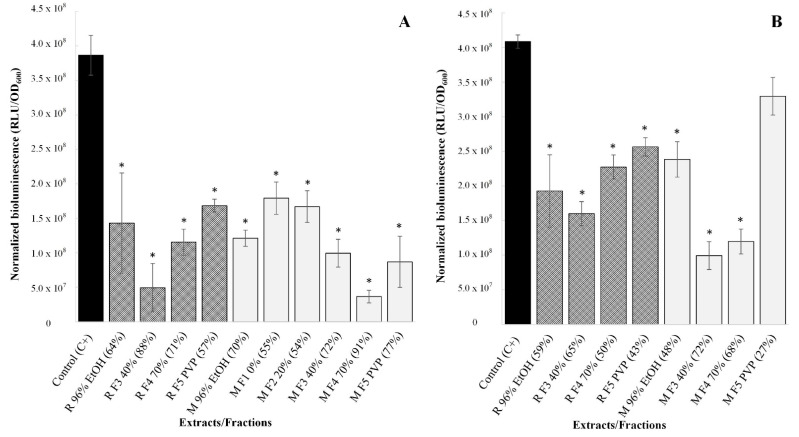
Bioluminescence reported after the addition of *V. harveyi* MM30 to 5% of SM obtained from non-treated control (C+) or to 5% of SMs obtained after treating *C. jejuni* NCTC 11168 with different preparations in sub-inhibitory concentrations ((**A**): 0.25× MIC; (**B**): 15.625 mg/L). Average bioluminescence in RLU with deducted background and normalized to OD_600_ are presented ± standard deviations. In addition, the bioluminescence reduction rate is presented in brackets. * represents statistically significant values.

**Figure 3 antibiotics-11-01220-f003:**
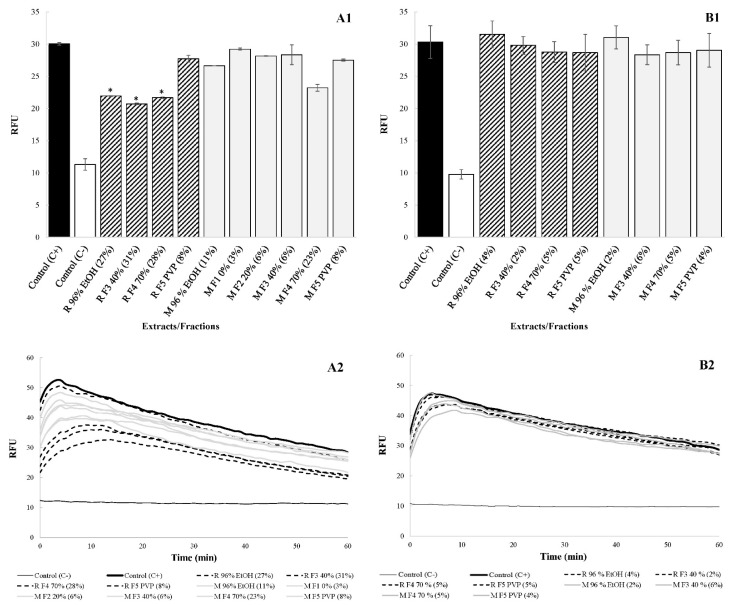
Influence of the extracts and their fractions on *C. jejuni* NCTC 11168 membrane integrity ((**A1**/**A2**): preparations tested at 0.25× MICs; (**B1**/**B2**): preparations tested in the same concentration, i.e., 15.625 mg/L). An impact on membrane integrity in RFU is presented ± standard deviations. In addition, the decrease in membrane integrity is presented in brackets. * represents statistically significant values.

**Table 1 antibiotics-11-01220-t001:** Fractions obtained from “Mattmark” and “Rosavine” ethanolic extracts and their main compounds.

No.	Label	Solvent	Main Compounds/Compound Groups
1	M/R F1 0%	MW, 0% methanol	Rich in gallic acid (Figure 1A)
2	M/R F2 20%	20% methanol	Rich in salidroside (Figure 1B)
3	M/R F3 40%	40% methanol	Rich in salidroside, rosavins, and PACs (Figure 1C)
4	M/R F4 70%	70% methanol	Rich in rosavins and flavonoids (Figure 1D)
5	M/R F5 PVP	50% methanol	Rich in salidroside and rosavins (contains almost no PACs or flavonoids) (Figure 1E)

**Table 2 antibiotics-11-01220-t002:** Determined MICs of ethanolic extracts and their fractions for *C. jejuni* NCTC 11168; *C. jejuni* 11168∆*luxS*.

Extracts/Fractions	*C. jejuni* NCTC 11168	*C. jejuni* 11168∆*luxS*
	MIC (mg/L)	
	**“Mattmark” Plant Material**	
M 96% EtOH	125	125
M F1 0%	250	250
M F2 20%	125	125
M F3 40%	62.5	62.5
M F4 70%	125	125
M F5 PVP	250	250
	**“Rosavine” Plant Material**	
R 96% EtOH	500	250
R F1 0%	>1000	1000
R F2 20%	>1000	500
R F3 40%	250	62.5
R F4 70%	250	125
R F5 PVP	1000	500

## Data Availability

The data are available in the Appendix A.

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
