# Peer review of "Rhodiola rosea Reduces Intercellular Signaling in Campylobacter jejuni"

_antibiotics, 2022, doi:10.3390/antibiotics11091220_

Round 1
Reviewer 1 Report
The authors obtained fractions of the initial ethanolic extracts rich in most important bioactive compounds from “Mattmark” and “Rosavine”, including salidroside, rosavins, proanthocyanidins (PACs), and flavonoids. moreover, authors determinate MICs, intercellular signaling by AI-2 bioassay, and Membrane Integrity. The manuscript is well presented. However, in my opinion there are several points that should be clarified.
1. In line 51, the author claimed AI-2- mediated signaling can be considered a potential target for controlling that pathogen in the introduction. Also they conclude that PACs and flavonoids are crucial compound groups responsible for the antimicrobial activity of R. rosea roots and rhizomes in C. 28 jejuni in the abstract. The point is I think the title and conclusion in the abstract can’t fully reflect the full content of this manuscript. Otherwise, do the authors suggest the antimicrobial activity of R. rosea was through reduced intercellular signaling, but not toxicity or killing it directly? If so, they should confirm it with C. jejuni 11168∆luxS.
2. As we all know, PACs and flavonoids have antimicrobial activity, what was the novelty of the antimicrobial activity of PACs and flavonoids of R. rosea roots and rhizomes? And what is the potential mechanism?
Reviewer 2 Report
In this article, the authors describe the protocol for the extraction of several compounds' groups from R. rosea. In particular, they prepared efficiently ethanolic extracts characterizing the composition by UV-HPLC-MS. Known compounds have been identified in the different fractions as well as the antimicrobial activity has been evaluated. The results appeared promising also for further investigation on this plant species and are clearly presented.
The manuscript is suitable for publication in Antibiotics
The paper is original and improves the research of antimicrobial plant extracts. Since interesting activities have been highlighted, this preliminary study could be followed by a new research aimed to isolate and structurally characterize all components of the ethanolic extract. The authors use a fluid and correct english language giving a step-by-step speech. One note: Table S1 should be more clearly when fragmentation patterns are shown
Round 2
Reviewer 1 Report
this work can be accepted for publishing in present form.